# Evaluation of Biologics ACE2/Ang(1–7) Encapsulated in Plant Cells for FDA Approval: Safety and Toxicology Studies

**DOI:** 10.3390/pharmaceutics17010012

**Published:** 2024-12-25

**Authors:** Henry Daniell, Geetanjali Wakade, Smruti K. Nair, Rahul Singh, Steven A. Emanuel, Barry Brock, Kenneth B. Margulies

**Affiliations:** 1Department of Basic & Translational Sciences, School of Dental Medicine, University of Pennsylvania, Philadelphia, PA 19104, USA; gwakade@upenn.edu (G.W.); smrutin@upenn.edu (S.K.N.); rsingh02@upenn.edu (R.S.); 2Department of Medicine, Perelman School of Medicine, University of Pennsylvania, Philadelphia, PA 19104, USA

**Keywords:** biologic, oral drug delivery, Angiotensin Converting Enzyme 2, CMC, toxicology

## Abstract

**Background/Objectives:** For several decades, protein drugs (biologics) made in cell cultures have been delivered as sterile injections, decreasing their affordability and patient preference. Angiotensin Converting Enzyme 2 (ACE2) gum is the first engineered human blood protein expressed in plant cells approved by the FDA without the need for purification and is a cold-chain and noninvasive drug delivery. This biologic is currently being evaluated in human clinical studies to debulk SARS-CoV-2 in the oral cavity to reduce coronavirus infection/transmission (NCT 0543318). **Methods:** Chemistry, manufacturing, and control (CMC) studies for the ACE2/Ang(1–7) drug substances (DSs) and ACE2 gum drug product (DP) were conducted following USP guidelines. GLP-compliant toxicology studies were conducted on Sprague Dawley rats (*n* = 120; 15/sex/group) in four groups—placebo, low (1.6/1.0 mg), medium (3.2/2.0 mg), and high (8.3/5.0 mg) doses IP/kg/day. Oral gavage was performed twice daily for 14 days (the dosing phase) followed by the recovery phase (35 days). Plasma samples (*n* = 216) were analyzed for the product Ang(1–7) by ELISA. **Results:** The ACE2 protein was stable in the gum for at least up to 78 weeks. The toxicology study revealed the dose-related drug delivery to the plasma and increases in the AUC (56.6%) and Cmax (52.9%) after 28 high-dose gavages (95% C.I.), although this quantitation excludes exogenously delivered membrane-associated ACE2/Ang(1–7). Vital biomarkers and organs were not adversely affected despite the 10-fold higher absorption in the tissues, demonstrating the safety for the first in-human clinical trials of ACE2/Ang(1–7). The NOAEL observed in the rats was 2.5–7.5-fold higher than that of the anticipated efficacious therapeutic dose in humans for the treatment of cardiopulmonary disorders, and it was 314-fold higher than the NOAEL for topical delivery via chewing gum. **Conclusions:** This report lays the foundation for the regulatory process approval for noninvasive and affordable human biologic drugs bioencapsulated in plant cells.

## 1. Introduction

Scientific advances have resulted in the revolutionary development of biologics to alleviate complex health disorders as well as diseases. Biologics are gaining more scientific attention in various applications due to their higher specificities, lower adverse effects, tolerance, and safety [1]. The global biologic market is expected to reach USD 394.2 billion by 2026 from USD 265.0 billion in 2021 at a compound annual growth rate (CAGR) of 8.3% until 2026 [2]. The first recombinant therapeutic protein drug, insulin (Humulin^®^; 1982), was approved in 1982 by the FDA [3]. In the past four decades, the FDA has approved >239 biologics and their 380 drug variants.

However, >90% of the FDA-approved biologics are delivered via invasive drug administration approaches [4,5]. From a pharmacokinetic standpoint, the parenteral routes (subcutaneous SC, intramuscular IM, and intravenous IV) of drug delivery are near ideal due to their high bioavailability, rapid onset of action, and enhanced delivery to the target site [6,7]. However, a major limitation of parenteral administration is poor patient compliance/preference. These modes of delivery require frequent dosing due to their short circulating half-lives and rapid clearance from the systemic circulation [8]. A restrictive dose volume, bleb formation, and induration at the injection sites are some of the other limitations of the parenteral drug delivery platforms [9,10,11]. Another roadblock is the need for expensive technologies like fermentation, purification, and cold storage and transportation, which increase the production costs, making it unaffordable to a considerable percentage of the global population [4,12]. Owing to patient-related concerns and the logistics and costs pertaining to the development of injectable formulations, academic and industrial efforts are being channeled towards developing noninvasive strategies that address patient adherence to treatment and affordability.

Topical drug delivery facilitates the rapid onset of action at the targeted site with minimal adverse reactions. Recently, a plant-based biologic administered via adhesive patches for the treatment of peanut allergy completed its phase III trial and was found effective in toddlers aged 1–3 years [13]. Although chewing gums with small molecules have been used in drug delivery for decades [14,15,16,17,18], it is only recently that a biologic has been delivered using a chewing gum [19]. The CTB-ACE2 protein produced in edible lettuce and delivered in a chewing gum formulation was effective at debulking > 90% of the SARS-CoV-2 virus in COVID-19 patient saliva and swab ex vivo samples and has received FDA approval to be evaluated for its efficacy in decreasing the infection and transmission of SARS-CoV-2 in phase I/II clinical trials (NCT05433181) [12,19].

The challenges associated with oral drug delivery are, viz., an acidic pH, enzymes, mucus permeation, and peptidase metabolism in the gastrointestinal tract, and the large molecular size and high hydrophilicity limit its therapeutic application [20,21]. Although numerous oral formulations have been developed to circumvent these limitations [21,22,23,24], a lack of sustained release and low in vivo intestinal uptake are two of the drawbacks of these drug delivery approaches [25]. The challenges in oral drug delivery inspired the creation of the European Network on Understanding Gastrointestinal Absorption-Related Processes (UNGAP) (www.ungap.eu, accessed on 25 November 2024), with several hundred multidisciplinary investigators from 32 countries [26]. The major topics of discussion by the UNGAP group are on formulations to enhance drug absorption by the gastrointestinal tract and impact the gut microbiome.

Fortunately, the oral delivery of biologics bioencapsulated in plant cells offers a novel alternative solution that addresses many of these challenges. The oral delivery of Ara h proteins in peanut cells is effective at desensitizing individuals who are allergic to peanuts. Based on a successful phase III clinical trial [27,28], this drug (Palforzia) was approved by the FDA and is used in the clinic. While Ara h proteins are naturally produced in peanut cells, engineered human proteins in plant cells have not yet been approved by the FDA. The oral delivery of glucocerebrosidase expressed in carrot cells has been tested in human clinical trials to treat Gaucher’s disease, but the drug dose should be enhanced further to achieve therapeutic efficacy [29]. Nevertheless, this study provided the proof of concept and safety evaluation for the oral delivery of biologics made in plant cells. Numerous preclinical studies conducted with mice [30,31], rats [32,33], and dogs [34,35] are advancing clinical trials to treat different diseases. These studies show that bacteria-producing enzymes that disrupt the plant cell wall (by the cleavage of the ß-1,4 and 1–6 linkages) are located in the duodenum [36], and therefore changes observed in the gut microbiomes in different animal models do not significantly impact the release and absorption of biologics in the gut. After their release in the gut lumen, the absorption of biologics by gut epithelial cells is a major limitation, but this has been addressed by using protein fusion tags that have been previously approved by the FDA and that are cleaved after gut absorption by engineered furin cleavage sites; furin is ubiquitously present in most human cells [30,33,34,35,37]. In addition to gut epithelial absorption, biologics are also delivered directly to the circulation without tags through the gut–liver axis, as demonstrated in the aforementioned studies.

The significance of oral drug delivery is the reduction in the cost of production and the delivery of biologics. Recent studies show that the average cost to develop a new biological product is USD ~2.6 billion because of the complexity of the current production and invasive drug delivery methods [38]. Since 2015, >90% of FDA-approved biologics have been injectable drugs requiring prohibitively expensive production in fermentation systems, purification, cold chains for transportation/storage, and invasive injections [4]. A plant cell-based treatment for a peanut allergy drug (Palforzia) recently approved by the FDA [27,28] costs < 3% that of the average cost of newly developed biologics for various applications [4,39]. Such a reduction in cost is achieved through the elimination of expensive purification and fermentation techniques, making this an ideal, affordable therapeutic approach. This is a major technological improvement not yet achieved for an engineered human drug over seven decades.

The aforementioned advantages of plant cell-based oral or topical delivery bode well for developing a regulatory path for therapeutic proteins for regulatory (FDA) approval. In this study, we report the specific steps required to obtain FDA approval for two human blood proteins: ACE2 and Ang(1–7) (Figure 1). The first IND submitted for the oral delivery of ACE2/Ang(1–7) was not approved by the FDA due to limitations in the toxicology studies, including the small group size, inadequate histopathological studies, short duration of drug delivery, inadequate assessment of the endpoints, and lack of a recovery phase to monitor the reversibility of clinical signs. Based on guidance received from the FDA, these concerns were addressed in the current GLP-compliant rodent toxicology study (gross pathology, histopathology, ophthalmology, body weights, clinical chemistry analysis of liver and renal function tests, hematology, coagulation, metabolic syndrome tests, urinalysis), which resulted in the subsequent approval of IND 154897 and its current evaluation in a phase I/II clinical trial (NCT05433181). We also report a comprehensive chemistry, manufacturing, and control analysis, including the drug dose stability, intra-batch content uniformity, moisture content, and bioburden evaluation of the ACE2 gum drug product.

## 2. Materials and Methods

### 2.1. Chemistry, Manufacturing, and Control Studies for CTB-ACE2 and CTB-Ang(1–7) Drug Substances (DSs) and Drug Product (DP)—ACE2 Gum

CTB-ACE2 and marker-free CTB-Ang(1–7) transplastomic lettuce lines were created after codon optimization, as reported in previous publications [33,40,41,42]. Clinical-grade biomass for drug proteins CTB-ACE2 and CTB-Ang(1–7) was grown, harvested, and lyophilized at Fraunhofer USA, Inc. (Plymouth, MI, USA) (cGMP facility) as per Daniell et al. [12] The biomass for both drug proteins was ground aseptically, and clinical-grade ACE2 gum tablets were prepared as per Daniell et al. [19].

#### 2.1.1. Intactness Assessment of CTB-ACE2 and CTB-Ang(1–7) Drug Substances

An amount of 10 mg of ground plant powder was accurately weighed in an Eppendorf tube using a precision balance for 3 and 5 s ground samples. An amount of 500 µL of freshly prepared plant extraction buffer (PEB) (NaCl: 100 mM; Ethylenediaminetetracetic Acid (EDTA): pH 8.0, 10 mM; Tris: 8.0, 200 mM; Sodium Dodecyl Sulphate: 0.1%; Sucrose: 400 mM; Tween: 20 0.05%; Dithiothreitol: 100 mM; protease inhibitor cocktail: 1×; Phenylmethylsulfonyl Fluoride: 2 mM; ß-Mercaptoethhanol: 10 mM) was added to the Eppendorf tube containing the plant sample and incubated at 4 °C on vortex for 5 min. After incubation, all the samples were centrifuged for 5 min at 750× *g*. After centrifugation, the supernatant (SUP) from each sample was removed carefully. The volume for the SUP samples was recorded for each sample. All the SUP samples were stored in an ice bath for protein estimation by Bradford Assay. To the Eppendorf tube containing pellets, 300 µL of PEB was added and mixed properly. Pellet samples were incubated on vortex for 1 h at 4 °C. After incubation, all the pellet samples were sonicated for 10 s ON and 15 s OFF 6 times. The samples after sonication were designated as homogenates (HMGs) and stored in an ice bath for further quantitation. The SUP and HMG fractions were diluted using water at 1:10 and 1:5, respectively, depending upon the absorbance (Linear Absorbance Measurement range—0.1 to 0.5) for the Bradford protein assay measurement. Before loading onto a 96-well plate, all the samples were centrifuged for 10 min at 14,000 rpm to remove any suspended particles contributing to turbidity. The SUP and HMG for the 3 and 5 s ground samples were analyzed for their total protein content by Bradford assay [19,43] and the expression of CTB-ACE2 by Western blot.

#### 2.1.2. Stability Assessment of CTB-ACE2 and CTB-Ang(1–7) DSs and DP (ACE2 Gum)

After grinding, samples were stored in FDA-approved black Uline containers at USP controlled room temperature and protected from light for >12 months. Assessment of the dose quantitation of CTB-ACE2 and CTB-Ang(1–7) from the drug substances was performed at <1 month and after 12 months of storage at room temperature to compare the loss of CTB-ACE2 during extended periods of storage at room temperature. An amount of 10 mg of ground plant powder was accurately weighed in an Eppendorf tube using a precision balance for the lyophilized ground samples. An amount of 500 µL of freshly prepared plant extraction buffer (PEB) (NaCl: 100 mM; Ethylenediaminetetracetic Acid (EDTA): pH 8.0, 10 mM; Tris: 8.0 200 mM; Sodium Dodecyl Sulphate: 0.1%; Sucrose: 400 mM; Tween: 20 0.05%; Dithiothreitol: 100 mM; protease inhibitor cocktail: 1X; Phenylmethylsulfonyl Fluoride: 2 mM; ß-Mercaptoethhanol: 10 mM) was added to the Eppendorf tube containing the plant sample and incubated at 4 °C on vortex for 1 h. After incubation, all the extracted samples were sonicated for 10 **s** ON and 15 s OFF 6 times. The samples after sonication were stored in an ice bath for further quantitation. Samples were diluted using water at 1:20 depending upon the absorbance (Linear Absorbance Measurement range—0.1 to 0.5) for the Bradford protein assay measurement. Before loading onto a 96-well plate, all the samples were centrifuged for 10 min at 14,000 rpm to remove any suspended particles. The total protein contents in the samples were tested using the Bradford assay [19,43], and expression of CTB-ACE2 was tested by Western blot. Drug dose quantitation of CTB-ACE2 and CTB-Ang(1–7) for DSs at <1 month and after 12 months were evaluated to assess the stability of the drug proteins. ACE2 gum was quantified for a CTB ACE2 dose at 4, 12, 26, and 78 weeks, stored at room temperature and protected from light.

#### 2.1.3. Bioburden Assessment

The procedure for the bioburden assessment was performed as per USP <61> and <62> for the enumeration of the microbial load and the test for specific microbial pathogens. The Biosafety Cabinet (BSC) was pre-run and kept ready for use as per the digital indicator after disinfection with 70% IPA. Sterile trypticase soy agar (TSA) plates (Becton Dickinson) and sterile Sabouraud’s dextrose agar (SDA) plates (Oxoid) were used for the enumeration of the aerobic bacteria and yeasts/molds, respectively. A 1% sample concentration was prepared aseptically by suspending 100 mg of ground plant powder in 10 mL of sterile diluent saline (0.85% Sodium chloride) followed by serial dilutions (10^−1^, 10^−2^, 10^−3^, 10^−4^ and 10^−5^) for the enumeration of the microbial load. Samples were vortexed to ensure the homogeneity of the suspension. Each dilution was spread inoculated aseptically onto TSA and SDA agar plates in duplicates and kept face up for 5 min in BSC. All the plates were sealed with parafilm and placed face down in the respective incubators (TSA—37 °C; SDS—25 °C) for 5 days. Uninoculated sterile plates and plates inoculated with sterile saline only were incubated as negative controls. Plates were monitored daily, and results were recorded on the 5th day of incubation. Similarly, enrichment of the specific microorganisms *E. coli* and *Salmonella* spp. were performed as described in USP <62>, and the enriched samples were inoculated onto MacConkey’s agar and Xylose lysine decarboxylate agar, respectively, in duplicates. Plates were incubated along with negative control plates at 37 °C for 48 h. DSs [CTB-ACE2 and CTB-Ang(1–7)] were assessed for bioburden after grinding before ACE2 gum preparation. ACE2 gum tablets were assessed for bioburden test at 4, 12, 26, 52, and 78 weeks, stored at room temperature and protected from light.

#### 2.1.4. Moisture Content Analysis by Karl Fisher Coulometry

Moisture content was analyzed based on USP <921> Method I © (coulometry). An empty glass vial was used for performing the titration. An amount of 50.0 ± 2.0 mg of Standard HYDRANAL-Water Standard—KF Oven 150–160 °C (Cat. Code 34693, Honeywell Specialty Chemicals Seelze GmbH) powder was weighed, and the sample mass was recorded in the KF machine. Vials were capped with aluminum seals. The instrument (Titrando 860 KF Thermoprep and Titrando 851, Metrohm Inc., Plainsboro, NJ, USA) was turned ON. The temperature was allowed to come up to 150 °C and the pump flow was allowed to come up to 50 mL/min on the Titrando 860 KF Thermoprep. A constant drift in the range of 10.0 ± 5.0 µg/min was attained. Three empty vials for blank calibration were run followed by 2 standard repeat injections to obtain the moisture content in the acceptable range (5.0% ± 0.04%). An amount of 50 mg of the drug substance was weighed in triplicates and assessed on a KF coulometer for moisture content before ACE2 gum preparation. Moisture content of ACE2 gum tablets was evaluated at 4, 12, 26, 52, and 78 weeks, stored at room temperature and protected from light.

#### 2.1.5. Content Uniformity Testing for ACE2 Gum

The CTB-ACE2 dose was determined for within the same ACE2 gum tablets and between different tablets to ensure drug dose accuracy between the tablets using the Western blot assessment method.

#### 2.1.6. Potency Study for ACE2 Gum Using Pseudo-Virus Neutralization Assays

Potency studies were evaluated for the ACE2 gum tablets at 4, 12, 26, and 52 weeks, stored at room temperature and protected from light, using pseudo-virus neutralization assays.

### 2.2. Preclinical Toxicology Assessment

This study was performed in accordance with the FDA (Food and Drug Administration) and U.S. HSS (Department of Health and Human Services), as accepted by Regulatory Authorities throughout the European Union (MHLW), Japan, and other countries that attest to the OECD Mutual Acceptance of Data Agreement (OECD Principles of Good Laboratory Practice), United States CFR (Code of Federal Regulations), and GLP (Good Laboratory Practice) for Nonclinical Laboratory Studies: Title 21, Part 58.

#### 2.2.1. Animals and Husbandry

Details on the animals for testing are depicted below:
Species:RatStrain:Crl:CD(SD) Sprague Dawley ratCondition:Purpose-bred, naiveSource:Charles River Kingston, Stone Ridge, NYNumber of Males:81 (plus at least 8 alternates)Number of Females:81 (plus at least 8 alternates)Age at the Initiation of Dosing:6 to 9 weeksWeight at the Initiation of Dosing:175 to 350 g (males)/175 to 250 g (females)

Following randomization, animals were housed in groups (up to 3 animals of the same dosing group, same sex and/or untreated alternates together). Caging of the animals was conducted in polycarbonate cages containing appropriate bedding equipped with an automatic watering valve.

Cages were color-coded with cage cards indicating study, group, animal number(s), and sex. Housing was as described in the Guidebook for the Care and Use of Laboratory Animals (National Academy Press, Current Edition) and as specified in the USDA Animal Welfare Act (Parts 1, 2, 3, and 9 CFR). Animals were separated during designated procedures/activities or as required for monitoring and/or health purposes, as deemed appropriate by the Clinical Veterinarian and/or Study Director. Cages were arranged group-wise on the racks. Placebo animals were housed on a separate rack from the test article-treated animals. Animals were housed socially and were provided with items such as a hiding device and/or a chewing object for psychological/environmental enrichment, except when interrupted by study procedures/activities. Edible enrichment treats were offered throughout the study. The targeted conditions for the animal room environment were as follows: humidity—30–70%; light cycle—12 h light and 12 h dark (except during designated procedures); temperature—68 °F–79 °F (20 °C–26 °C); and ventilation—10 or more air changes per hour.

#### 2.2.2. Preparation of CTB-ACE2 and CTB-Ang(1–7) Dose Formulations for Oral Gavage

The dose formulation of the investigational products, CTB-ACE2 batch—1 (7.32 mg/g DW) and CTB-Ang(1–7) (4.14 mg/g DW), were prepared as described in Appendix A. Dose formulations were prepared as needed in phosphate-buffered saline and stirred for at least 30 min before dosing. The dosing formulations were protected from light and stirred continuously during dosing to ensure homogeneity. Also, all the formulations were used within approximately 2 h after preparation and orally gavaged to Sprague Dawley rats of the respective groups twice daily (8 h ± 1 h apart) from day 1 to day 14 using a syringe with an attached gavage cannula. The first day of dosing was designated as day 1.

#### 2.2.3. Toxicokinetic Assessment of CTB-ACE2 and CTB-Ang(1–7)

The potential toxicity of the fusion proteins CTB-ACE2/CTB-Ang(1–7) expressed in lettuce chloroplasts was tested in a 14-day repeat-oral-dose study with a 20-day recovery period to assess the potential reversibility of any findings in the rats to evaluate the safety and efficacy of the drug proteins. For the toxicokinetic study, four groups of Sprague Dawley rats were administered the placebo (3/sex) or mixed test agents, the LS-CTB-ACE2/LS-CTB-Ang(1–7) fusion proteins, via oral gavage twice daily for 14 days at doses of 1.6/1.0 mg IP/kg/day (low dose; 6/sex), 3.2/2.0 mg IP/Kg/day (medium dose; 6/sex), or 8.3/5.0 mg IP/Kg/day (high dose; 6/sex) (Appendix A). Animals designated for terminal euthanasia (10/sex/group) were euthanized on day 15 and animals designated for recovery euthanasia (5/sex/group) were euthanized on day 35 following a minimum of 20-day recovery. In-life assessments were performed as depicted in Appendix A.

These parameters included mortality, clinical observations, body weight and food consumption, ophthalmology, and clinical pathology parameters. A total of 216 samples were collected at 0, 2,4, 6, 10, and 24 h after 2 and 28 gavages to analyze the plasma Ang- (1–7) conc. at all timepoints (Figure 2) by ELISA kit (Cloud-Clone Corp., Texas, USA, Product code—CES85Hu, 96 tests), as per the manufacturer’s instructions. Samples were collected and processed for clinical pathology parameters (clinical chemistry—liver function, real function, and metabolic syndrome tests, hematology, coagulation, urinalysis), as per Appendix A.

Ophthalmology examination prior to in-life initiations was performed on all animals designated for potential assignment to the main and recovery study periods. Both eyes of each animal were examined by a veterinarian or veterinary ophthalmologist with appropriate training in this species using a hand-held slit lamp and indirect ophthalmoscope. A short-acting mydriatic solution was instilled into each eye to facilitate the ocular examinations. For gross pathology, the organs identified for weighing for all scheduled euthanasia animals were weighed at necropsy. Organ weights were not recorded for animals euthanized or found dead in poor condition or in extremis. In addition, the weights of paired organs were recorded. In cases of gross abnormalities, in addition to the combined weight, the weight of the individual organs of a pair was entered as a tissue comment. Organ weight as a percent of terminal body weight and as a percent of brain weight was calculated. For histopathology, representative samples of tissues were preserved in 10% neutral-buffered formalin, unless otherwise indicated.

A total of 120 animals (60 male and 60 female) were used in the main and recovery studies (dosing and recovery phases) that are included in Figure 1 and Appendix A. A total of 42 animals were used for the toxicokinetic study (21 males and 21 females), which are included in Appendix A. So, a total of 162 (81 male and 81 female) animals were used in the composite studies described in the Animals and Husbandry Section.

### 2.3. Parametric/Non-Parametric

Statistics and computerized systems:

The homogeneity of the group variances was assessed using Levene’s test. Group comparisons were performed using an overall one-way ANOVA F-test wherever the Kruskal–Wallis test was significant or where Levene’s test was not significant. Pairwise comparisons were performed using Dunnett’s or Dunn’s test if the overall F-test or Kruskal–Wallis test was found to be significant. Datasets with 2 groups were compared using Dunn’s test (equivalent to Wilcoxon Rank-Sum test in Nevis 2012 tables) or Dunnett’s test (equivalent to *t*-test in Nevis 2012 tables). Computerized systems and their versions for analysis of different parameters in the toxicology study performed at Charles River Laboratories are listed in Appendix A.

## 3. Results

Previously, we reported the codon optimization of both human genes ACE2 and Ang(1–7), the generation of transplastomic lines, biomass optimization, and the evaluation of the potency in the treatment of pulmonary hypertension upon oral delivery [32,33] or in trapping SARS-CoV-2 upon the topical delivery of ACE2 [12,19]. High-level expression is a basic requirement for the oral and topical delivery of biologics, as pointed out above for the oral delivery of glucocerebrosidase expressed in carrot cells. Therefore, to enhance the ACE2 and Ang(1–7) expression in lettuce chloroplasts, both human genes were codon-optimized as per the hierarchy of the most highly expressed chloroplast gene, psbA [40]. In the codon optimization process of the human ACE2 gene, among 805 codons, 481 codons, including 59 rare codons, were replaced. Both proteins were fused with CTB to facilitate delivery across the gut epithelium. The protein production parameters were also optimized for a higher yield [41]. In this study, we evaluated the chemistry, manufacturing, and control, intactness of the plant cells to protect biologics in the stomach, stability, uniformity, moisture content, and bioburden.

The first step in the evaluation of any new drug is seeking approval from the FDA to proceed with human clinical trials. This requires drug product characterization (CMC studies) and nonclinical efficacy, disposition, and safety studies. Typically, for a CDER-regulated new drug, GLP-compliant, multiple-dose in vivo safety studies in a rodent and non-rodent model are required to demonstrate safety and define a safe starting dose in humans. This manuscript provides a detailed description of our GLP-compliant, IND-enabling rodent toxicology study. In addition, we provide safety data on the oral delivery of ACE2/Ang(1–7) in healthy dogs. These data satisfied the FDA requirements for preclinical safety assessment.

### 3.1. Chemistry, Manufacturing, and Control (CMC) Data for CTB-ACE2 and CTB-Ang(1–7) Drug Substances (DSs) and Drug Product (DP)—ACE2 Gum

CMC studies are crucial in the drug development process, which involves the manufacturing of drug substances and drug products, process controls, analytical methods, and stability assessments to ensure the identity, purity, and activity of different batches to release them for preclinical and clinical studies [44]. The dose, quality, consistency, stability (dose, moisture content, bioburden, etc.), and potency of the investigational drugs are important criteria evaluated during drug development [45]. Both the DSs and DP (ACE2 gum) were characterized, and manufacturing controls were included in the study as per USP (United States Pharmacopoeia) guidelines in accordance with the FDA, and critical quality attributes (CQAs) were identified for both the DSs and DP.

### 3.2. Stability Assessment of CTB-ACE2 and CTB-Ang(1–7) Drug Substances

The rigorous optimization of the lettuce plant growth conditions and nutrients to achieve a higher expression of CTB-ACE2 and CTB-Ang(1–7) were conducted as reported previously [12]. The stabilities of the CTB-ACE2 and CTB-Ang(1–7) in the plant biomass were evaluated by comparing the drug protein levels quantified after one year of storage at USP controlled room temperature, protected from light. The CTB-ACE2 drug dose of the plant samples harvested on different days showed maximums of 17.2 and 17.9 mg/g DW and the CTB-Ang-(1–7) drug dose showed maximums of 15.6 and 15.4 mg/g DW when analyzed less than one month and after 1 year of storage, respectively, suggesting a <3.5% reduction in the drug dose, which was statistically insignificant and within the batch-to-batch experimental variations (Figure 3A). Similar to previously reported stability data (24–31 months) of other orally delivered therapeutic proteins within lettuce plant cells [31,34,46,47], the CTB-ACE2 and CTB-Ang(1–7) remained stable during long-term storage at the USP controlled temperature. The regression analysis results of the stability testing (drug dose vs. time) are well within the limits of the 95% CI in accordance with the principles detailed in the ICH guidance Q1A(R2) Stability Testing of New Drug Substances and Products, demonstrating long-term storage at room temperature (https://www.fda.gov/media/71722/download, accessed on 23 November 2023).

### 3.3. Intactness Assessment of Plant Cells Expressing Protein Drugs

The intactness of plant cells is required for the protection of protein drugs from acids and enzymes. Therefore, the grinding time was optimized to facilitate the optimal release of the drug protein bioencapsulated within the plant cells, depending on the mode of delivery. The grinding of the lyophilized plant biomass for 3, 6, 9, and 12 *s* resulted in particle size reductions, which were measured by the increase in the drug protein release, of 5.6, 11.4, 20.6, and 25.6%, respectively, with the increasing grinding time (Figure 3B,D). The grinding of the lyophilized plant leaves for 3, 6, 9, and 12 *s* revealed the gradual release of the CTB-ACE2 protein in the supernatant. The maximum release of CTB-ACE2 (25.6%) was observed at 12 *s* and 5.6% at 3 *s*, which are desirable for topical and oral drug delivery, respectively (Figure 3B). The lyophilized CTB-Ang(1–7) plant biomass was ground for two durations—3 and 5 *s* (Figure 3C). The intactness of the plant cells was 85% and 72.4% in the 3 *s* and 5 *s* ground samples, respectively (Figure 3C). The controlled grinding of the plant biomass for 3 *s*, where 95% of the plant cell was intact, was chosen for the oral delivery of ACE2/Ang(1–7) for the toxicology studies in rats. For the topical delivery of CTB-ACE2 to debulk the SARS-CoV-2 viral load, the 12 *s* ground DS was used for the rapid release of the drug protein from the ACE2 gum [19].

### 3.4. Bioburden and Moisture Content as per USP Guidelines

The USP specifications and observed parameters for orally delivered drugs are listed, including the purity, bioburden, and moisture content of the drug substance. The drug substance and placebo plant powder containing a mixture of proteins (Generally Regarded as Safe) passed the bioburden test with no aerobic bacteria or yeasts/molds on the trypticase soy agar plates and Sabouraud’s agar plates, respectively, after an incubation of 5 days at 30 °C (USP <61>). Also, the samples tested negative for the growth of Salmonella and *E. coli* spp., as per USP <62> (Table 1). An analysis of the moisture content is an essential test in drug development process, as it influences the quality, stability, shelf life, and safety of the drug protein. The moisture contents for both CTB-ACE2 and CTB Ang(1–7) were 5.5 ± 0.05 and 5.8 ± 0.03%, respectively. Based on the absence of viable microorganisms and their stable moisture contents, these batches were used for the toxicology studies and the manufacturing of the ACE2 gum tablets. Thus, the USP compliance of the DSs for the bioburden and moisture content demonstrated the safety and stability of the drug proteins for future clinical trials.

### 3.5. Release Criteria for ACE2 Gum

To evaluate the safety (bioburden and moisture content), efficacy (drug dose, potency), and consistency (uniformity) of the drug product, the gum tablets were assessed for their critical quality attributes (CQAs) at specified time intervals, viz., 4, 12, 26, 52, and 78 weeks. The CTB-ACE2 in the chewing gum was stable up to 78 weeks with marginal degradation (0.4%) at USP controlled room temperature. The potency of the ACE2 gum was 100% at 0.92 μg CTB-ACE2 (pseudo-virus neutralization assay) (*p* < 0.0001) up to 52 weeks. The moisture contents of the CTB-ACE2 gum and placebo were minimal (1.25%). The uniformity in the drug dose was evaluated in different gum tablets or different segments of the same gum tablet. As shown in Figure 1, the content uniformity was 99.8% within a single tablet and 99.5% between different tablets of the same batch, suggesting the uniformity of the drug substance. The ACE2 gum and placebo showed no microbial growth at 4, 15, 26, 52, and 78 weeks. Thus, the CQA assessment revealed that the DP is stable up to 78 weeks at USP controlled room temperature protected from light (Table 2).

### 3.6. In Vivo Safety Assessment of CTB-ACE2 and CTB-Ang(1–7)

The in vivo safety of the CTB-ACE2/Ang(1–7) was assessed in a multiple-dose rat study conducted in accordance with FDA guidelines. The GLP-compliant study (Figure 2) followed a repeat-dosing regimen across a range of doses (0, 1.6/1.0 mg (low dose), 3.2/2.0 mg (medium dose), and 8.3/5.0 (high) mg IP/Kg/day) for 14 days followed by a recovery phase (day 35 evaluation). The doses were selected based on previous studies on the restoration of cardiopulmonary health in several animal disease models [32,33] and considering multiple factors, including the anticipated starting dose in humans, achieving a maximum feasible dose, the expression levels of ACE2 and Ang(1–7) in plant cells, and the integrity of the lyophilized plant cells in the dosing solutions. Daily doses of 1.6/1.0 mg/kg, 3.2/2.0 mg/kg, and 8.3/5.0 mg/kg of ACE2/Ang(1–7) for this study were based on a pharmacologically active low dose of ACE2/Ang(1–7) [33] and the maximum feasible high dose of ACE2/Ang(1–7). The low dose is six times higher on a mg/kg basis than the anticipated starting human clinical dose. Therefore, the study design used four groups: the placebo, low-, medium-, and high-dose groups. There must be sufficient protein (plant powder) present in the dosing solutions to prevent the osmotic lysis of plant cells. Because the total plant powder in group 2 was below this threshold, extra placebo plant powder was added, equivalent to that of groups 1, 2, and 3. For the high-dose group 4, the dose of ACE2/Ang(1–7) was based on the maximum amount of plant protein that could be gavaged twice daily with a maximum dose volume of 20 mL/kg.

#### 3.6.1. Clinical Observations

All rats gavaged with the placebo, low-, medium-, and high-dose mixes of CTB-ACE2/Ang(1–7) were observed for clinical abnormalities, viz., in respiration, breathing sounds, hunched posture, erected or stained fur, and decreased activity in any of the animals (Table 3 and Appendix A). There were no abnormal clinical observations noted during this study. Incidental findings like stained fur and skin scabs were not related to the CTB-ACE2/Ang(1–7) and are common background findings for this age and strain of rat.

The individual body weights and food consumption were recorded once every week in all the animals from the dosing and recovery studies. The body weights of the male rats continued to increase from 262 g on day 1 to 521 g on day 35, similar to those of the control groups fed the placebo. Similarly, the body weights were observed to increase from 207.3 g on day 1 to 316 g on day 35 for all the female rats with respect to the controls (Table 3, Appendix A).

The body weight gains were not significantly different for the treatment groups as compared to the controls for the male (250 ± 50 g) and female (100 ± 10 g) rats until the recovery period relative to the start date (Table 3, Appendix A). The food consumption patterns of all the treatment groups for the male (~30 g) and female rats (~25 g) were observed to be consistent with respect to the control groups during the dosing and recovery periods relative to the start date (Table 3, Appendix A). Thus, the oral CTB-ACE2/Ang(1–7) showed normal trends in the body weights or body weight gains for all the treatment groups after 2 and 28 gavages. All the fluctuations among the individual and mean values, regardless of their statistical significance, were considered sporadic, consistent with the biologic variation, and/or negligible in magnitude, and not related to the drug protein.

#### 3.6.2. Toxicokinetics

The samples (*n* = 216) collected at 0, 2, 4, 6, 10, and 24 h on both days 1 and 14 and the plasma Ang(1–7), a downstream metabolite of ACE2, were analyzed (Figure 4A) by ELISA. The average times for the drug to reach the maximum plasma concentration (Tmax) for the treated groups 2, 3, and 4 on day 1 were relatively slow compared to on day 14 (10.66 and 6.05 h, respectively). This suggests that the steady state of the plasma Ang(1–7) was attained by day 14 with 28 gavages. The presence of detectable Ang(1–7) at the 0 h timepoint (pregavage) and in the placebo group samples reflects the presence of the circulating endogenous Ang(1–7) in healthy rats. The pregavage average plasma Ang(1–7) concentration before the gavage across all the groups was found to be 509 ± 15 pg/mL. The percent increase in the plasma Ang(1–7) concentration was calculated based on the pregavage value (Figure 4A). The medium-dose group showed lower concentrations of plasma Ang(1–7). For each of the dosage levels, the average plasma levels were increased after 28 gavages (14 days) compared with day 1 (2 gavages). We observed dose-dependent increases in the Ang(1–7) concentrations across the three groups (95% C.I) (Figure 4A). The Cmax values for the low-, medium-, and high-dose groups on day 14 were observed to be 4.5, 27, and 53% higher than those on day 1 (Figure 4B). The AUClast values for all three groups—the low-, medium-, and high-dose groups—were 8.5, 22.8, and 56.5% higher than those on day 1 (Figure 4C). These results indicate that a repeat-dosing regimen of 28 gavages increased the Ang(1–7) drug concentration in the plasma compared to the two gavages administered on day 1. The higher plasma concentration observed in all the groups after the repeat-dose administration of the CTB-ACE2/CTB-Ang(1–7) fusion proteins is a result of the efficient oral delivery of the plant protein drugs, which is a requirement in dose-dependent toxicology and efficacy studies.

#### 3.6.3. Gross Pathology

Two female rats each from the high- and medium-dose mixes were euthanized on day 3 and day 7 due to severe gavage injury, and the deaths were serendipitous and unrelated to the test agents. All the other animals survived until the scheduled necropsies. No test article-related differences in the values of the mean organ weights were noted. There were no isolated organ weight values that were statistically different from their respective controls. There were no patterns, trends, or correlating data to suggest that these differences were test article-related, and they were considered incidental and unrelated to the administration of the mixed test agents, CTB-ACE2/Ang(1–7). A complete gross pathological examination was performed on all the animals and the organ weights were recorded, as specified in Table 4A,B and Table 5A,B.

The percentage difference in the body weights after gavaging the control, low-dose, medium-dose, and high-dose mixes was consistent across all the groups relative to their original body weights on day 15 and day 35. The terminal body weights of the males and females on day 15 after two gavage studies and on day 35 (during the recovery phase) for all the groups, viz., the low-, medium-, and high-dose mixes, were found to be statistically insignificant (*p* > 0.05; ANOVA and Dunnet’s test) with respect to their controls with a few exceptions unrelated to the test articles (Table 4A,B and Table 5A,B).

The individual organ weights for the male and female rats (male rats—brain, epididymis, adrenal, prostate, thyroid/parathyroid, heart, kidney, liver, spleen, testis; female rats—thymus, ovary, uterus/cervix) noted on day 15 and day 35 (the recovery phase) exhibited statistically insignificant differences (*p* > 0.05; ANOVA and Dunnet’s test) with respect to the un-transfected controls (Table 4A,B and Table 5A,B). These observations indicated that there was no evidence of edema, lesions, or inflammation in the individual organ weights of either the male or female rats.

#### 3.6.4. Histopathology

Microscopic examinations were performed at the time of terminal necropsy on animals from all the groups—the placebo, low-, medium-, and high-dose groups—in the oral gavage (10/sex/group) and recovery (5/sex/group) phases. Observations were made on day 15 and day 35. The organ systems analyzed were the integumentary, musculoskeletal, lymphatic, endocrine, nervous, respiratory, cardiovascular, gastrointestinal, reproductive, and urinary systems (Table 6). The evaluations revealed no remarkable test article-related microscopic observations or other aberrant findings in the tissues of the vital organs of the animals in the test article-fed groups compared to the placebo (Table 6). The organs of the digestive system, viz., the stomach, large intestine (cecum, colon, and rectum), small intestine (duodenum, jejunum, and ileum), and rectum, were examined. A potential test article-related finding was noted as focal erosion in the pylorus of the glandular stomach in one female gavaged with the high dose of ACE2/Ang(1–7). There was no toxicity of the orally delivered drug protein in the organs despite the higher drug protein accumulation observed in the tissues [33].

#### 3.6.5. Ophthalmology

No adverse ocular findings were observed in the rats fed the drug proteins. Also, no drug-related abnormalities in the eyelids, conjunctiva, lenses, cornea, cortex anterior, or posterior segments of the eyes were revealed through indirect ophthalmoscopy or bio-microscopy for all animals fed the low-, medium-, high-dose formulations, including the placebo control. Ophthalmic examinations were not performed at the end of the recovery period because no treatment-related findings were present during the last week of dosing. This confirms that the drug proteins do not have observable ocular toxicity (Table 7).

#### 3.6.6. Hematology

The complete blood cell count (CBC) is an important set of biomarkers to assess inflammation, and inflammation-related indexes can identify the potential toxicity hazards of drugs or pharmaceutical substances [48]. The results of the screening for hematologic toxicity are shown in Figure 5. All the CBC parameters, viz., the hematocrit (male—42–44%; female—40–42%), hemoglobin concentration (male—14–15 g/dL; female—13.6–15 g/dL), erythrocyte count (male—6.9–7.9 × 10^6^ g/dL; female—6.7–7.4 × 10^6^ g/dL ), total and differential leukocyte counts (male—7.0–12 × 10^3^ µL; female—6.4–12 × 10^3^ µL), mean corpuscular hemoglobin (male—19.3–20.7 pg; female—9.3–20.6 pg), mean corpuscular volume (male—57.9–61.9 fL; female—57.2–60.5 fL), and mean corpuscular hemoglobin concentration (male—32.7–34.0 g/dL; female—33.9–34.7 g/dL) were statistically insignificant at a 95% CI for rats gavaged with the low-, medium-, and high-dose mixes, including the placebo control, after 2 and 28 gavages in the dosing and recovery phases and were normally distributed in their reference intervals for young healthy Sprague Dawley rats (Table 3, Figure 5) [49].

#### 3.6.7. Clinical Chemistry

##### Liver Function Test

A standard battery of clinical chemistry endpoints was assessed. As shown in Figure 6 and Figure 7, the values determined for the AST (male and female rats: 80–100 U/L), ALT (male and female rats: 30–40 U/L), AP (male rats—200–270 U/L; females—130–153 U/L), and GGT (male and female rats—1–1.5 U/L) for all the groups, including the placebo control, after 2 and 28 gavages in the dosing and recovery phases were found to be distributed in the normal reference values for young healthy Sprague Dawley rats, suggesting no adverse liver effects upon drug administration [50,51]. Similarly, the Total Bilirubin (TBIL) (0.008–0.04 mg/dL), total protein (male and female rats—5.5–5.8 g/dL), albumin (male and female rats—3–4 g/dL), and globulin (1.5–1.7 g/dL), which are indicators of nutritional status, immune function, cholestasis, and liver disorders, were not statistically altered after 2 and 28 gavages of any of the dose formulations compared to the control groups. These values were in the normal reference interval range for healthy Sprague Dawley rats (Table 3, Figure 6). The remaining differences in the clinical chemistry parameters, regardless of statistical significance, were not considered related to the CTB-ACE2 or CTB-Ang(1–7) administration based on their small magnitude, inconsistent direction, and general overlap of individual values within the range of the control values at a 95% CI.

##### Renal Function Test

As shown in Figure 7, the administration of increasing doses of LS-CTB-ACE2/CTB-Ang(1–7) did not alter the renal function in the animals in the dosing phase and at post-recovery on day 35. The mean values of the renal function parameters—BUN (12.2 mg/dL ± 1.1), creatinine (0.2 mg/dL), calcium (10.4 mg/dL ± 0.1), sodium (142.3 mEq/L ± 0.6), potassium (4.7 mEq/L ± 0.3), chloride (101.7 mEq/L ± 0.4), phosphate (9 mg/dL ± 0.6), and muscle damage—creatine kinase (410 U/L ± 81.8)—were found to be within the normal reference ranges in rats at a 95% CI [52].

#### 3.6.8. Urinalysis

Urine samples were collected after overnight fasting and subjected to an analysis of their volume, specific gravity, and pH at terminal euthanasia (dosing phase) and at the end of the recovery phase. The parameters did not change significantly between the dosing and recovery phases. The volume, specific gravity, and pH were 7.5 ± 1.4 mL, 1.0, and 7 ± 0.4, respectively [52,53]. No test fusion protein Ls-CTB-ACE2/CTB-Ang (1–7)-related changes were detected at any dose level at a 95% CI (Table 3, Figure 8A).

#### 3.6.9. Coagulation

Coagulation tests, including those for the prothrombin time (PT), active partial thromboplastin time (PTT), and fibrinogen, were performed on the plasma samples to determine whether the test fusion proteins CTB-ACE2/CTB-Ang(1–7) affected the blood clotting in the rats. These in vitro tests measured the time elapsed from the activation of the coagulation cascade until the formation of fibrin at various timepoints. No CTB-ACE2/CTB-Ang(1–7)-related changes in the coagulation parameters were noted at any dose level compared to the control group. The mean values for the PT, PTT, and FIB in all the animals across all the dose groups were 16 s ± 0.16, 14.6 s ± 0.02, and 299 mg/dL ± 30.4, respectively. These values are well within the normal range [54] and did not change significantly between the dosing and recovery phases at a 95% CI (Table 3, Figure 8B).

#### 3.6.10. Metabolic Syndrome Tests

Physiological biomarkers like glucose, cholesterol, and triglycerides play an important role in predicting the risk of underlying metabolic disorders like cardiovascular disease and diabetes [55,56]. The measurement of these parameters revealed no significant differences across any of the groups or between day 15 (the dosing phase) and day 35 (the recovery phase). The mean values for the glucose, cholesterol, and triglycerides in the animals at all the dose levels were 132.5 mg/dL ± 6.1, 69.3 mg/dL ± 4.6, and 32.2 mg/dL ± 3.1, respectively, which is within the normal reference ranges at a 95% CI (Table 3, Figure 8C) [57].

## 4. Discussion

There are essential evaluations central to the IND application comprising details on the investigative product (IP) manufacturing process, systemic toxicity evaluation, and IP disposition (ADME/PK) properties. Due to the dispositional nature of protein therapeutics, the ADME properties are less relevant when compared to small-molecule therapeutics. With regards to manufacturing, inter- and intra-lot variations in the stability, uniformity, and potency, to name a few, are crucial to achieve uniform drug administration during clinical applications. The identification of dose-limiting toxicities, such as ocular toxicity hazards, and the estimation of a safe starting dose in the first clinical trial, are achieved by a rigorous in vivo toxicity study.

Different batches of the DSs CTB-ACE2 and CTB-Ang(1–7) were extensively analyzed as per USP testing procedures, and the CQA parameters were defined, which are integral parts of an IND application. The freeze drying of the transgenic lettuce helps in maintaining the protein stability for extended storage durations at ambient temperature. With a negligible loss of protein (< 3.5%), the protein drug remained stable after a storage period of 1 year. Additionally, we demonstrated extended stability for at least 31 months for several human blood proteins expressed in chloroplasts grown and lyophilized under similar conditions [30,33,34,42,47]. Intactness evaluations of plant cells are important to facilitate oral drug delivery, as most protein drugs are degraded in the harsh gastric environment [58,59,60,61,62]. Hence, the grinding time of the drug substance was optimized to facilitate the oral and topical drug delivery method. The grind time was optimized to balance the release of enough protein for topical administration, which is intuitively reduced when a more systemic distribution is required via oral means. The oral route mandates that the plant cells are intact, which is conferred through bioencapsulation, which protects them from premature degradation in the stomach until they are disrupted by enzymes secreted by commensal bacteria. Unlike orally delivered ACE2/Ang(1–7), the efficient topical delivery of these drug proteins occurs from broken plant cells for rapid release in the oral cavity. This was evident in in vitro human chewing simulation experiments, in which most of the drug protein was released during the initial chewing time, with a linear increase in the drug release up to 20 min [19].

The delivery of ACE2 via chewing gum is not intended to be systemic. Indeed, plant cells are ground for a longer duration to facilitate rapid release in the oral cavity to trap the coronavirus. Proteins released from broken plant cells are digested in the stomach by acids and enzymes. The toxicology studies reported here were conducted to obtain FDA approval for the phase I/II clinical trials for the oral delivery of ACE2/Ang(1–7) to treat COVID-19 patients, or for the topical delivery of ACE2 gum to decrease the infection and transmission of SARS-CoV-2 (NCT05433181). Several previous studies have tested the role of the systemic delivery of ACE2/Ang(1–7) in clinics to treat RAS-associated metabolic diseases or COVID-19 infection that significantly impacts the RAS pathway [63,64,65,66,67,68,69,70]. The dose-dependent attenuation of monocrotaline-induced pulmonary arterial hypertension with the oral delivery of CTB-ACE2/Ang(1–7) has been investigated in animal models. The key findings include the absence of toxicity, the arrest of the disease progression with a decrease in the right ventricular (RV) hypertrophy, RV systolic pressure, total pulmonary resistance, and pulmonary artery remodeling [32,33].

Microbial Enumeration Tests, Ph. Eur. 2.6.12/USP <61>, and “Microbiological Examination of Nonsterile Products: Tests for Specified Microorganisms”, Ph. Eur. 2.6.13/USP <62>), are crucial in pharmaceutical industries to ensure the safety and quality of a drug substance or drug product, as well as the starting material to monitor hygiene during the manufacturing process [71]. The drug substances and ACE2 gum passed the bioburden test for the microbial load and specific human pathogens (USP <61> and USP <62>), making it safe for clinical applications. Lyophilization increases the shelf-life and facilitates the storage and transport of the drug substances/product at ambient temperature [72]. An increase in the moisture activates the proteolytic enzymes and increases the microbial load, adversely affecting the stability of the drug protein [73]. The moisture contents for both the drug substances and drug product were minimal, improving their shelf life, quality, and stability for extended storage periods at USP controlled temperature.

The toxicology study described in this paper suggests that the NOAEL (no-observed-adverse-effect level) is the high dose (8.3/5 mg/kg BW) and supports the need for repeated dosing to observe a dose-dependent increase in the plasma Ang(1–7) by this orally delivered bioencapsulated protein drug formulation. The dose-dependent increase observed in the plasma Ang(1–7) is due to the efficient degradation of the plant cell wall and drug absorption in the gut. The NOAEL dose is 3.6 (ACE2) and 1.1-fold [Ang(1–7)] higher than the predicted effective dose to treat cardiopulmonary disorders in humans [33]. A well-tolerated high dose of oral ACE2/Ang(1–7) with no adverse events was a critical element for advancing the fusion protein ACE2/Ang(1–7) to the clinic. The volume of evidence substantiating the role of ACE2 in RAS dysregulation and the pathogenesis of COVID-19 [74,75,76] propelled several human clinical trials using rhACE2 [66,67,68,69] and synthetic Ang(1–7) as therapeutic agents. By reversing Angiotensin II (Ang II)-mediated systemic inflammation by increasing the impact of cardioprotective Ang(1–7), ACE2 alleviated pulmonary arterial hypertension (PAH) and Acute Respiratory Distress Syndrome (ARDS)-induced systemic inflammation [65,66]. The role of Ang(1–7) in mitigating the proinflammatory effect of Ang II is evident in various cardiovascular animal models [32,33,77] and human clinical studies [67,68,78]. Previous studies with multiple animal models captured the cardioprotective effect of oral ACE2/Ang(1–7). In rats, oral ACE2/Ang(1–7) attenuated the PAH and improved the cardiopulmonary function via reducing the ventricular hypertrophy, vascular remodeling, and increasing the ejection [32,33].

Moreover, as reported in Daniell et al. [12,19], the ACE2 gum efficiently debulked the SARS-CoV-2 virus count by >95% in COVID-19 patient saliva and nasopharyngeal swab samples by the direct binding of the spike protein to soluble ACE2, as measured by microbubbling (N antigen), a RAPID (spike protein) assay, and qPCR, demonstrating both virus trapping and the blocking of cellular entry. These studies highlight the strength of delivering plant viral trap proteins via chewing gum to neutralize pathogens in the oral cavity and on throat surfaces, thereby reducing the infection and transmission of SARS-CoV-2. The IND 154897 approval of the ACE2 gum led to the phase I/II placebo-controlled, double-blind clinical trial to evaluate the impact of the gum in reducing the viral load in saliva. The clinical trial study design consists of a study duration of 4 days with 13 gums in total, with four gums each on days 1–3. An unstimulated whole saliva will be collected before eating, drinking, or brushing teeth, and the subjects will then chew the CTB-ACE2 gum/placebo gum for 10 min and immediately provide a post-treatment sample of 2–5 mL. Subsequently, the subjects will be asked to chew the test CTB-ACE2/placebo gum for 10 min at home in the AM, before lunch, in late afternoon, and before sleeping on days 1, 2, and 3. Saliva samples will be collected before eating, drinking, or brushing teeth and after treatment with the ACE2 gum. On the final day 4, samples will be collected once in the morning before brushing, eating, or drinking. The viral load will be quantified by qPCR or N or spike quantitation. The clinical endpoints include the Primary Endpoint Reduction in the viral load in saliva, measured by qPCR quantitation before and after chewing the gum, as discussed in Daniell et al. [19]. This directly assesses the gum’s effectiveness in debulking the virus, which is crucial for reducing transmission. The secondary endpoints are the safety and tolerability, monitoring the adverse events and overall tolerability of the chewing gum in the participants. Analysis of the viral load kinetics consists of the evaluation of the duration of viral load reduction in the saliva over multiple days of gum usage per day. Analysis of the symptom severity consists of assessing the impact of the reduction in the viral load on the severity and progression of COVID-19 symptoms using a daily symptom score.

The recent approval of Rybelsus^®^ (an oral semaglutide) by the FDA and EMA for type 2 diabetes (2019) [79] and octreotide (Mycapssa) for acromegaly (2020) [80] suggests progress in the field of oral protein drug delivery. The scientific progression over the last 30 years for oral Rybelsus^®^ has reduced the cost of the drug delivery significantly; however, the oral API manufacturing cost has increased relative to that of injection-based products, making no progress in the overall cost of treatment, which is a major concern for patient access [81]. The affordable approach using the oral delivery of protein drugs using edible plant cells has unique advantages. The US FDA recently approved the plant cell-based oral delivery of a biologic to induce tolerance against peanut allergy (Palforzia). The annual cost of this drug (360 capsules with peanut powder) is USD ~2500, which is significantly less than the average annual cost (USD ~85,508) of the fermentation-based injectable drug [4,5,39]. The topical delivery of this drug to children through patches was also recently approved by the FDA [82]. The recently launched antiviral drug Paxlovid costs USD 1400 for a three-day treatment (Pfizer sold this drug for USD 8 billion in 2023) [83]. Therefore, there is a great need for affordable antiviral drugs. FDA authorization to proceed with the first human blood protein encapsulated in plant cells—Angiotensin converting enzyme 2 (ACE2)—in chewing gums for the neutralization of the SARS-CoV-2 in saliva to limit transmission and self-infection (IND 154807, NCT05433181) is one example of this. Based on the regulatory approvals of plant cell-made drugs by the FDA, we anticipate that the topical delivery of viral trap protein will have transformative potential for improving accessibility, affordability, and compliance.

In addition to being the first human blood protein to be permitted to proceed by the FDA, this is also the first time that a fusion tag protein has been authorized to proceed. The role of CTB is that of a transmucosal carrier that transports the protein drug through the GI tract and releases it after binding to the GM1 receptors expressed on the epithelial cells lining the small intestine [40]. This is followed by the degradation of the CTB pentamer and is not detected in the circulation. Moreover, CTB is already FDA-approved and has been in clinical use for several decades [84]. Several studies show the immune tolerance and immune suppressing effect of CTB fusion proteins, which makes them ideal candidates for fusion tag proteins for protein delivery [42,44,45]. This concept has been applied to several protein therapeutics, like oral insulin, and they are promising, leading to advancements in this technology to alleviate a range of metabolic disorders. The ACE2 gum drug product underwent rigorous characterizations at the pre-determined timepoints following FDA recommendations. The absence of bioburden and specific pathogens and the minimal moisture content ensured the safety and stability with a significantly longer shelf life than parenteral drugs. In addition, the negligible degradation of the drug protein and 100% neutralization in the pseudo-virus assays achieved by the drug product stored for 78 weeks at ambient temperature further confirms the feasibility and affordability of this potent drug delivery platform. Moreover, each 2 g ACE2 gum tablet containing 750 μg of CTB-ACE2 and <1 μg of the drug protein achieved 100% neutralization. Also, with a dose 314-fold lower than that of the NOAEL dose in the topical ACE2 gum, no toxic effects are anticipated.

The stability of the CTB-ACE2 in the chewing gum and its functional efficacy in neutralizing (100%) pseudo-viruses even stored at ambient temperature up to 78 weeks clearly indicate its robustness under varying storage and transportation conditions. In the pseudo-virus neutralization assay, we found that 50 mg of the CTB-ACE2 gum was sufficient for 100% neutralization (Table 2), while the gum formulation employed is a 2000 mg dose. This suggests that the drug product (ACE2 gum) presented in this study is 40-fold higher than the required efficacy. The 40-fold higher efficacy of the Ace2 gum further justifies its robustness and translational ability in the clinic. In the planned randomized, placebo-controlled phase I/II trial, 260 CTB-ACE2 and 260 placebo chewing gums are required to conduct a trial with a total of 40 COVID-19 patients for 4 days of treatment. Although the stability and functional efficacy of the CTB-ACE2 in the gum is evaluated up to 78 weeks in this study, we anticipate CTB-ACE2 stability and functional efficacy for even longer periods of time because a recently reported FRIL gum [85] prepared following the same method used for the CTB-ACE2 gum is functionally stable up to 823 days (117.5 weeks). Overall, the reported long-term stability of several different plant cell-based protein therapeutics in chewing gum stored at ambient temperature [85,86] justifies its broader application and eventually transportability across the globe, particularly in resource-limited countries where cold storage and transportation is a problem. The importance of this technology became more relevant when ~19 million dosages of COVID-19 vaccines were discarded in Africa due to the unavailability of cold storage and transportation facilities [5,87].

The lower bioavailability of orally delivered drugs and the need for higher dosages to attain the optimum functional efficacy are among the major challenges in therapeutic protein drug development. For instance, comparable semaglutide efficacies were observed between the oral 50 mg/QD and the 2.4 mg/week subcutaneous semaglutide in the treatment of diabetes and obesity, the former being 21-fold higher [88,89]. Oral delivery offers an advantage in delivering drugs efficiently to the liver via the hepatic portal vasculature, which would otherwise be reduced to 25–30% when administered parentally [90]. Significant glucodynamic effects were observed with the oral enteric insulin capsule despite no detectable rise in the serum insulin concentration with an increased dose [91]. In a recent study, oral insulin regulated the blood sugar levels without causing hypoglycemia, supporting the optimal pharmacodynamic effect of exogenous plant protein drugs [31]. Thus, for orally delivered drugs, measuring local pharmacodynamic events is also equally important, along with the PK-PD parameters [90].

Rodent models do not fully capture the complexity of human diseases, and operational challenges require significant infrastructure and collaboration. So, the dose estimation/exchange between species during research, experiments, and clinical trials requires careful consideration. Allometric scaling that considers the differences in the body surface area, which is associated with animal weight, could assist scientists to exchange doses between species during research, experiments, and clinical trials [92]. Several topically delivered plant cell-derived viral trap/antiviral products translated from preclinical findings to clinical outcomes in human trials have been reported. The viral trap polymer Iota-carrageenan obtained from red algae [93] topically delivered in the form of a nasal spray protected healthcare workers from the infection/transmission of SARS-CoV-2 from COVID-19 patients [94]. A plant extract of Echinaceae/Salvia delivered topically in the form of lozenges successfully reduced the viral load up to 96% after 4 days of treatment in a patient with a sore throat, but adverse events were observed in patients because of the high doses employed (16800 mg per day) [95,96]. On the basis of the in vitro evaluated potency (100% neutralization of pseudo-virus by 50 mg of ACE2 gum), we are anticipating better clinical outcomes for the drug product (ACE2 gum) developed in this study for preventing SARS-CoV-2 re-infection in the same patient or its transmission to others. Since the ACE2 gum is efficacious at very low doses, minimum adverse events are expected in the planned clinical phase I/II trials. Moreover, the ACE2 gum should have advantages because chewing gum as a delivery vehicle is preferable compared to lozenges because of the slow and prolonged release of the drug [5]. However, the actual clinical outcome of the ACE2 gum will only be confirmed by the planned phase I/II clinical trials.

## 5. Conclusions

For several decades, biologics have been made in cell cultures and have been delivered as sterile injections, decreasing their affordability and patient preference while compromising their utility in many settings. The lack of vaccines for several viruses, low vaccination rates, waning immunity, and viral transmission after vaccination underscore the need to reduce viral loads at their transmission sites. Oral virus transmission is several orders of magnitude higher than nasal transmission. Current antiviral drugs like Paxlovid are unaffordable and not available for a large majority of the global population. Therefore, in this chemistry, manufacturing, and control study, we showed that the ACE2 protein is stable in the gum for at least 78 weeks when stored at ambient temperature. The ACE2 dose is uniform (99.8%) within a single tablet and between different tablets of the same batch (99.5%). In the GLP-compliant toxicology studies, the NOAEL observed in the rats was 314-fold higher for the topical delivery of ACE2 via chewing gum. Angiotensin Converting Enzyme 2 (ACE2) is the first engineered human blood protein expressed in plant cells approved by the FDA for evaluation in human clinical trials, without the need for prohibitively expensive fermentation, purification, cold-chain, and invasive drug delivery. This biologic is currently being evaluated in human clinical studies to debulk SARS-CoV-2 in the oral cavity to reduce coronavirus infection/transmission (NCT 0543318). Recognizing that the US FDA recently approved the plant cell-based oral delivery of biologics to induce tolerance against peanut allergies, the required regulatory pathways are in place to exploit the reduced costs and favorable implementation attributes of orally available, plant-derived therapeutics for delivering ACE2/Ang(1–7). These studies also advance the prospects for diverse plant-derived therapeutics more generally.

## 6. Patents

A complete list of the patents of HD is available at the following Scholar GPS and Google Scholar links: https://scholargps.com/scholars/82094026790000/henry-daniell (accessed on 16 December 2024) and https://scholar.google.com/citations?user=7sow4jwAAAAJ (accessed on 16 December 2024).

## Figures and Tables

**Figure 1 pharmaceutics-17-00012-f001:**
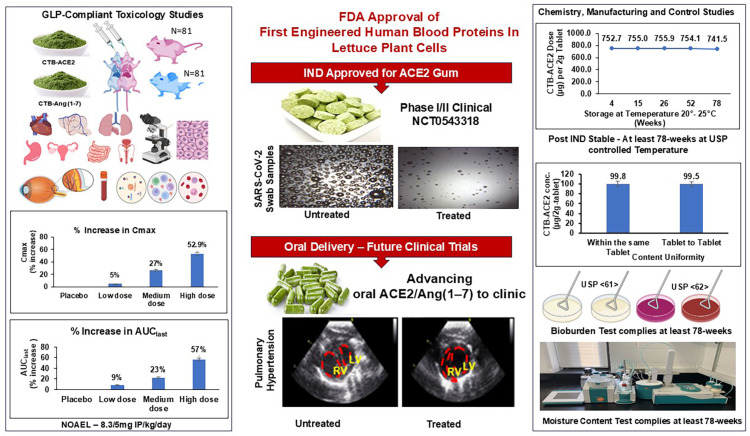
FDA-compliant CTB-ACE2/Ang(1–7) drug substances and product eliminates fermentation, purification, cold-chain, and invasive drug delivery methods. Left panel—GLP-complaint toxicology studies performed for a total of 162 animals with toxicokinetic assessment. Middle panel—topical delivery of IND-approved ACE2 gum to debulk SARS-CoV2 and oral delivery of ACE2/Ang(1–7) to clinic. Right panel—chemistry, manufacturing, and control studies of drug product—ACE2 gum. RV: right ventricle; LV: left ventricle; NOAEL: no-observed-adverse-effect level; ACE2: Angiotensin Converting Enzyme 2.

**Figure 2 pharmaceutics-17-00012-f002:**
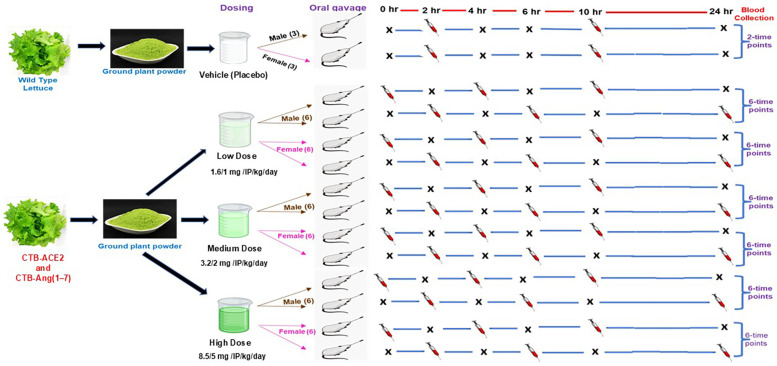
Toxicokinetic study design: CTB-ACE2 and CTB-Ang(1–7) proteins expressed in lettuce were orally gavaged twice daily for 14 days to male (*n* = 21) and female (*n* = 21) rats in placebo, low-, medium-, and high-dose groups. The plasma samples were collected at 2, 4, 6, 10, and 24 h on day 1 and day 14 and analyzed for the toxicokinetic and pharmacodynamic parameters.

**Figure 3 pharmaceutics-17-00012-f003:**
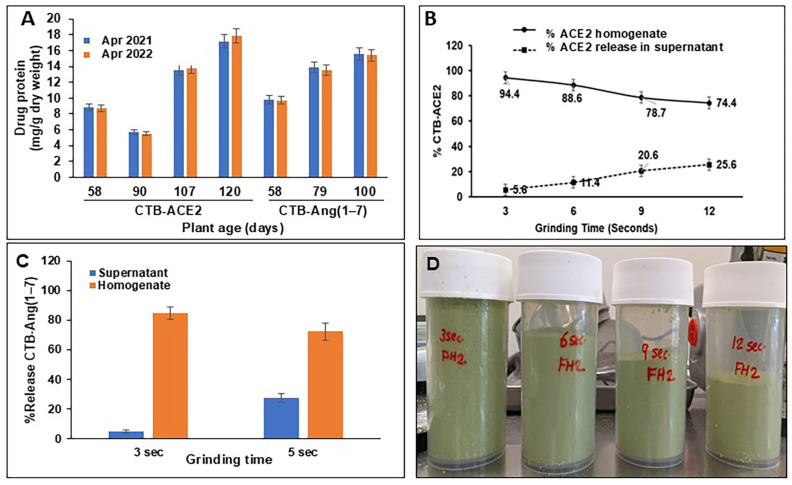
Chemistry, manufacturing, and control data of drug substances (DSs) CTB-ACE2 and CTB-Ang(1–7) stored at USP controlled temperature. (**A**) Stability of DSs assessed at <1 month and after 1 year of storage at USP controlled temperature in FDA-approved black containers protected from light. (**B**) Effect of grinding time on optimal release of drug protein CTB-ACE2 for oral (3 s) and topical (12 s) delivery. (**C**) Effect of grinding time on optimal release of drug protein CTB-Ang (1–7) for oral delivery (3 s). (**D**) Grinding time (3, 6, 9, 12 s)-dependent reduction in particle size.

**Figure 4 pharmaceutics-17-00012-f004:**
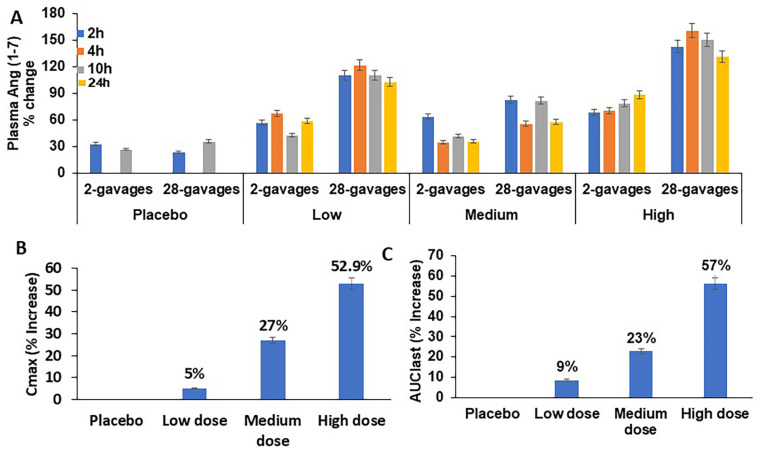
Toxicokinetic analysis of plasma Ang(1–7) concentrations. (**A**) Comparative plasma Ang-(1–7) concentrations in low-, medium-, and high-dose groups after 2 gavages (0 h and 8 h) and 28 gavages at 0, 2, 4, 10, and 24 h timepoints. Data presented as % change (95% CI) w.r.t pregavage plasma Ang-(1–7). (**B**) Percentage increase in C_max_ after 28 gavages (14 days) vs. 2 gavages (day 1) across placebo, low-, medium-, and high-dose groups. (**C**) Percentage increase in AUC_last_ after 28 gavages (14 days) vs. 2 gavages (day 1) across placebo, low-, medium-, and high-dose groups. Data presented as % increase (95% CI).

**Figure 5 pharmaceutics-17-00012-f005:**
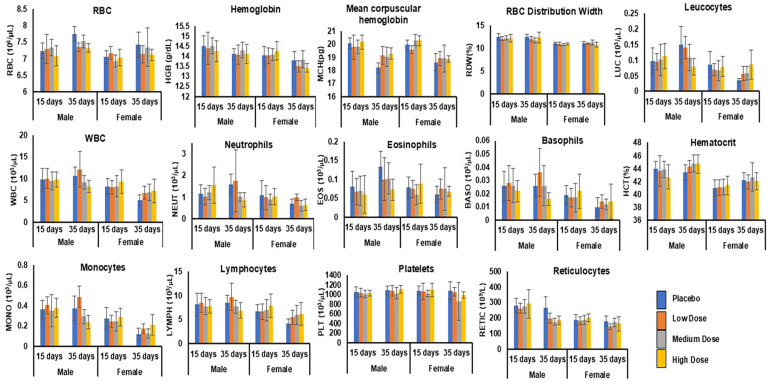
Hematology Index. Complete blood counts were measured in male (*n* = 15) and female (*n* = 15) rats for placebo, low-, medium-, and high-dose groups after 28 gavages at the end of the dosing phase (day 15) followed by the recovery phase (day 35). ANOVA with Dunnett’s method showed that the low, medium, and high doses did not produce more significant differences in the CBC parameters than those of the placebo.

**Figure 6 pharmaceutics-17-00012-f006:**
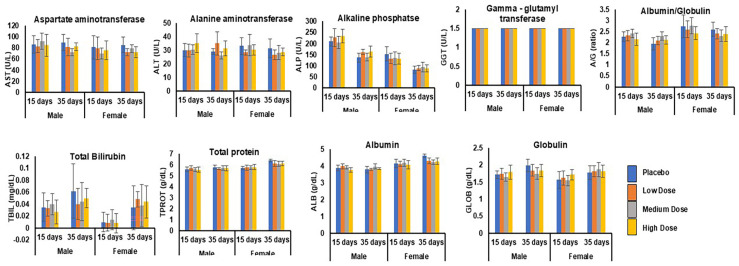
Liver function test. Liver enzymes were measured in male (*n* = 15) and female (*n* = 15) rats for the placebo, low-, medium-, and high-dose groups after 28 gavages at the end of the dosing phase (day 15) followed by the recovery phase (day 35). ANOVA with Dunnett’s method showed that the low, medium, and high doses did not produce more significant differences in the liver function test parameters than those of the placebo control.

**Figure 7 pharmaceutics-17-00012-f007:**
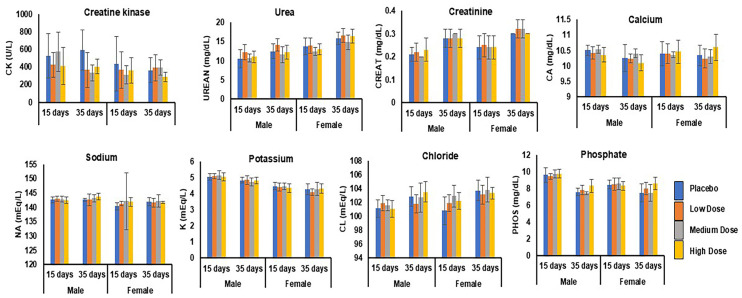
Renal function test. Renal enzymes were measured in male (*n* = 15) and female (*n* = 15) rats for placebo, low-, medium-, and high-dose groups after 28 gavages at the end of the dosing phase (day 15) followed by the recovery phase (day 35). ANOVA with Dunnett’s method showed that the low, medium, and high doses did not produce more significant differences in the renal function test parameters than those of the placebo control.

**Figure 8 pharmaceutics-17-00012-f008:**
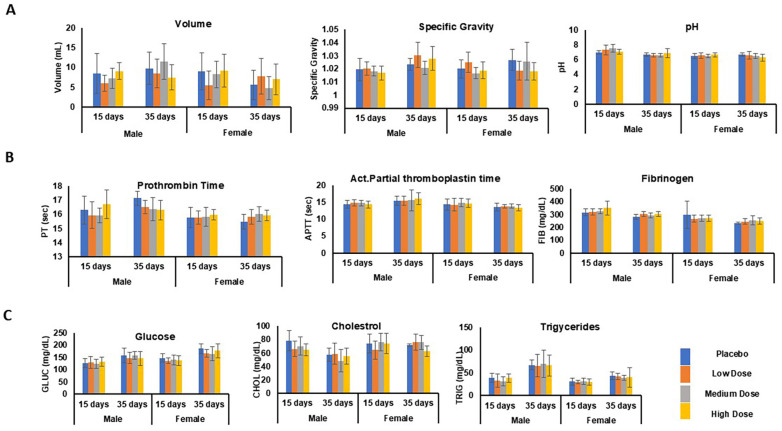
(**A**) Urinalysis (Urine volume, specific gravity and pH), (**B**) coagulation (Prothrombin Time, Act. Partial thromboplastin time, fibrinogen), and (**C**) metabolic syndrome test (glucose, cholesterol and triglyceride) parameters were measured in male (*n* = 15) and female (*n* = 15F) rats for the placebo, low-, medium-, and high-dose groups after 28 gavages at the end of the dosing phase (day 15) followed by the recovery phase (day 35). ANOVA with Dunnett’s method showed that the low, medium, and high doses did not produce more significant differences in the urinalysis, coagulation, and metabolic syndrome test parameters than those of the placebo control.

**Table 1 pharmaceutics-17-00012-t001:** Drug substance characterization for CTB-ACE2 and CTB-Ang(1–7) as per United States Pharmacopeia guidelines.

Drug Substance	CTB-ACE2	CTB-Ang(1–7)
Age	17 weeks, 14 days	16 months
Appearance—light- to dark-green powder	Pass	Pass
Total aerobic microbial count, 10^3^ cfu/g	0	0
Total yeast, 10^3^ cfu/g	0	0
Total molds, 10^4^ cfu/g	0	0
Absence of *Salmonella*	Absent	Absent
Absence of *Escherichia coli*	Absent	Absent
Assay (mg/g Dry wt.)	18.37 ± 0.89	15.4 ± 0.9
Moisture content	5.5%	5.8%

**Table 2 pharmaceutics-17-00012-t002:** Quality control criteria for ACE2 gum: bioburden, moisture content, stability, and potency (50 mg plant powder—18.8 mg/g DW ACE2) after storage of gum at ambient temperature for 4, 12, 26, 52, and 78 weeks.

Specifications	4 Weeks	12 Weeks	26 Weeks	52 Weeks	78 Weeks
Total aerobic microbial count, <10^3^ CFU/g	0	0	0	0	0
Total yeast count, <10^3^ CFU/g	0	0	0	0	0
Total mold count, <10^3^ CFU/g	0	0	0	0	0
Absence of *Salmonella*	Absent	Absent	Absent	Absent	Absent
Absence of *Escherichia coli*	Absent	Absent	Absent	Absent	Absent
Drug dose—250–1000 µg CTB-ACE2 per 2 g dry weight of gum tablet	752.6 ± 3.4 µg	755 ± 3.1 µg	755.9 ± 2.3 µg	754.1 ± 2.6 µg	741.5 ± 8.9 µg
Moisture content—<10%	1.2	1.24	1.25	1.3	1.33
Potency—80–110% neutralization at 50 mg ACE2 gum	100%	100%	100%	100%	100%

**Table 3 pharmaceutics-17-00012-t003:** Summary of parameters evaluated in the GLP-compliant rodent toxicology study.

Test	Parameters	Results	Reference
Hematology	RBC, Hb, MCH, RBW, LEUC, WBC, NEUT, EOS, BASO, HEMAT, MONO, LYMPHO, PLAT, RETIC	No statistically significant differences observed across dose groups compared to control animals.	Figure 5
Liver function	AST, ALT, ALP, GGT, TBIL, TIPRO, ALB, GLOB, A/G		Figure 6
Renal function	CK, UREA, CREAT, CA, PHOS, NA, K, CL.		Figure 7
Urinalysis	VOL, sp. gravity, pH		Figure 8A
Coagulation	PT, PTT, FIB		Figure 8B
Metabolic syndrome test	GLUC, CHOL, TRIG		Figure 8C
Body weights/body weight gain/food consumption	Weight measurement, food consumption in g		Appendix A
Gross pathology	Full battery of organs and tissues of all animals	No abnormalities detected.	Table 4 and Table 5
Histopathology			Table 6
Ophthalmology	Eyelids, conjunctiva, cornea, lens, iris, cortex, nucleus, indirect ophthalmoscopy, biomicroscopic examinations		Table 7
Clinical observations	Respiratory rate, breathing, posture, fur, activity	No abnormalities detected.	Appendix A

**Table 4 pharmaceutics-17-00012-t004:** (**A**) Gross pathology: organ weights of male rats on day 15. (**B**) Gross pathology: organ weights of male rats on day 35.

(**A**)					
**Sex: Male Rat** **Body/Organs**	**Weight (g) and % Difference**	**Control**	**Low Dose**	**Medium Dose**	**High Dose**
Original Body Weight	Mean	264.6	262.3	262.3	269.3
Terminal Body [G]	Mean	351.5	340.9	342.8	350.6
	%Diff ^a^	26.7	24.4	25.0	24.8
Brain [G]	Mean	2.0041	1.9748	1.9921	1.9582
	%Diff	-	−1.4620	−0.5988	−2.2903
Epididymis [G]	Mean	0.7873	0.7877	0.7666	0.7764
	%Diff	-	0.0508	−2.6292	−1.3845
Gland, Adrenal [G]	Mean	0.05786	0.05175	0.05435	0.052
	%Diff	-	−10.5	−6.0	−9.4
Gland, Pituitary [G]	Mean	0.01225	0.01098	0.01209	0.01273
	%Diff	-	−10.36735	−1.30612	3.91837
Gland, Prostate [G]	Mean	0.6615	0.7103	0.6787	0.6408
	%Diff	-	7.3772	2.6002	−3.1293
Thyroid/Parathyroid [G]	Mean	0.02235	0.02090	0.02084	0.01977
	%Diff	-	−6.48770	−6.756	−11.543
Heart [G]	Mean	1.3533	1.3433	1.35	1.27
	%Diff	-	−0.74	0.33	−5.7
Kidney [G]	Mean	2.5699	2.5686	2.4617	2.4654
	%Diff	-	−0.0506	−4.2103	−4.0663
Liver [G]	Mean	10.43	10.4256	10.2	10.50
	%Diff	-	−0.1035	−2.2604	0.6602
Spleen [G]	Mean	0.7858	0.7347	0.7894	0.7910
	%Diff	-	−6.50	0.45	0.66
Testis [G]	Mean	3.2570	2.9544	3.1996	3.1390
	%Diff	-	−9.29	−1.76	−3.62
Thymus [G]	Mean	0.5277	0.4706	0.5240	0.5391
	%Diff	-	−10.82	−0.70	2.16
(**B**)					
**Sex: Male Rat** **Body/Organs**	**Weight (g) and % Difference**	**Control**	**Low Dose**	**Medium Dose**	**High Dose**
Original Body Weight	Mean	264.6	262.3	262.3	269.3
Terminal Body [G]	Mean	445.6	483.6	465.4	469.8
	%Diff ^a^	68.4	84.4	77.4	74.5
Brain Weight [G]	Mean	2.0512	2.0706	2.0748	2.1232
	%Diff ^a^	-	0.9458	1.1505	3.5101
Epididymis [G]	Mean	1.0636	1.1960	1.0742	1.1946
	%Diff	-	12.4483	0.9966	12.3167
Gland, Adrenal [G]	Mean	0.05968	0.07038	0.06108	0.07020
	%Diff	0.00831	0.00481	0.00768	0.01486
Gland, Pituitary [G]	Mean	0.01332	0.01588	0.01388	0.01442
	%Diff	-	19.21922	4.20420	8.25826
Gland, Prostate [G]	Mean	0.8154	1.0784	0.9630	0.9886
	%Diff	-	32.2541	18.1015	21.2411
Thyroid/Parathyroid [G]	Mean	0.02654	0.02652	0.02366	0.02280
	%Diff	-	−0.07536	−10.85154	−14.09194
Heart [G]	Mean	1.4710	1.5446	1.5538	1.5792
	%Diff	-	5.0034	5.6288	7.3555
Kidney [G]	Mean	2.9388	2.9740	3.0762	3.0624
	%Diff	-	1.1978	4.6754	4.2058
Liver [G]	Mean	12.8086	14.3502	13.2068	13.1266
	%Diff	-	12.0357	3.1088	2.4827
Spleen [G]	Mean	0.9742	0.9222	0.9046	0.8174
	%Diff	-	−5.3377	−7.1443	−16.0953
Testis [G]	Mean	3.1518	3.6024	3.2626	3.5478
	%Diff	-	14.2966	3.5155	12.5642
Thymus [G]	Mean	0.5102	0.4798	0.5662	0.4630
	%Diff	-	−5.9584	10.9761	−9.2513

[G]—ANOVA and Dunnet’s: *p* > 0.05—(ns)*;*
^a^ % Difference in the terminal body weight was calculated based on the original body weight. The % difference for the organ weight was calculated with respect to the control.

**Table 5 pharmaceutics-17-00012-t005:** (**A**) Gross pathology: organ weights of female rats on day 15. (**B**) Gross pathology: organ weights of female rats on day 35.

(**A**)					
**Sex: Female Rat** **Body/Organs**	**Weight (g) and % Difference**	**Control**	**Low Dose**	**Medium Dose**	**High Dose**
Original Body Weight	Mean	205.9	201.5	201.5	207.3
Terminal Body [G]	Mean	247.2	232.4	240.8	245.6
	%Diff ^a^	20.1	15.3	19.5	18.5
Brain [G]	Mean	1.88	1.86	1.89	1.89
	%Diff	-	−1.0710	0.4719	0.6732
Gland, Adrenal [G]	Mean	0.06737	0.06959	0.06589	0.06988
	%Diff	-	3.29524	−2.19847	3.72240
Gland, Pituitary [G]	Mean	0.015	0.015	0.015	0.015
	%Diff	-	−2.27716	−5.44350	−2.40729
Thyroid/Parathyroid [G]	Mean	0.01646	0.01672	0.01863	0.01828
	%Diff	-	1.57959	13.20373	11.04361
Heart [G]	Mean	1.02	0.97	1.01	1.03
	%Diff	-	−4.8668	−1.0750	1.1990
Kidney [G]	Mean	1.88	1.79	1.83	1.83
	%Diff	-	−4.6767	−2.6135	−2.3235
Liver [G]	Mean	7.5775	7.4375	7.5021	7.6938
	%Diff	-	−1.8476	−0.9949	1.5345
Ovary [G]	Mean	0.0902	0.0910	0.0996	0.0958
	%Diff	-	0.8869	10.3720	6.1838
Spleen [G]	Mean	0.5747	0.5769	0.5737	0.5967
	%Diff	-	0.3828	−0.1798	3.8223
Thymus [G]	Mean	0.5813	0.5211	0.5647	0.6487
	%Diff	-	−10.3561	−2.8614	11.5890
Uterus/Cervix [G]	Mean	0.5512	0.5227	0.4920	0.6197
	%Diff	-	−5.1705	−10.7402	12.4214
(**B**)					
**Sex: Female Rat** **Body/Organs**	**Weight (g) and % Difference**	**Control**	**Low Dose**	**Medium Dose**	**High Dose**
Original Body Weight	Mean	205.9	201.5	201.5	207.3
Terminal Body [G]	Mean	288.4	295.0	280.0	298.2
	%Diff ^a^	40.1	46.4	39.0	43.8
Brain Weight [G]	Mean	1.9774	1.9784	1.9508	1.9702
	%Diff	-	0.0506	−1.3452	−0.3641
Gland, Adrenal [G]	Mean	0.07008	0.07348	0.06602	0.07878
	%Diff	-	4.85160	−5.79338	12.41438
Gland, Pituitary [G]	Mean	0.01750	0.01696	0.01640	0.01738
	%Diff	-	−3.08571	−6.28571	−0.68571
Thyroid/Parathyroid [G]	Mean	0.01808	0.02090	0.02420	0.01866
	%Diff	-	15.59735	33.84956	3.20796
Heart [G]	Mean	1.1000	1.1160	0.9948	1.1102
	%Diff	-	1.4545	−9.5636	0.9273
Kidney [G]	Mean	1.8836	1.9534	1.9060	1.9436
	%Diff	-	3.7057	1.1892	3.1854
Liver [G]	Mean	8.3656	8.6844	7.7492	8.3342
	%Diff	-	3.8108	−7.3683	−0.3753
Ovary [G]	Mean	0.0998	0.1076	0.0984	0.1058
	%Diff	-	7.8156	−1.4028	6.0120
Spleen [G]	Mean	0.5928	0.6336	0.6162	0.7262
	%Diff	-	6.8826	3.9474	22.5034
Thymus [G]	Mean	0.5154	0.4606	0.5362	0.5540
	%Diff	-	−10.6325	4.0357	7.4893
Uterus/Cervix (g)—[G]	Mean	0.6672	0.5628	0.7350	0.6720
	%Diff	-	−15.6475	10.1619	0.7194

[G]—ANOVA and Dunnet’s: *p* > 0.05—(ns); ^a^ % Difference in the terminal body weight was calculated based on the original body weight. The % difference for the organ weight was calculated with respect to the control.

**Table 6 pharmaceutics-17-00012-t006:** Histopathological findings of male and female rats on day 15 and day 35.

Organs Systems (Male and Female—*n* = 120)		Histopathological Observations in All Groups
Integumentary	Skin	No visible lesions
Musculoskeletal	Bone marrow, sternum, bone, femur, bone, sternum, joint, femorotibial, muscle, skeletal	
Lymphatic	Galt, mandibular, mesenteric, spleen, mediastinal	Minimal increase in plasma cell, cellularity, no visible lesions
Endocrine	Adrenal, harderian, mammary, parathyroid, thymus, pituitary, salivary, mandibular, thyroid, prostate, seminal vesicle	No aberrant craniopharyngeal structures, inflammation, hypertrophy, acinar cells, cyst, ultimobranchial lesions
Nervous	Brain, nerves—optic, sciatic, spinal cord	No visible lesions
	Eye	No visible lesions, retina folds
Respiratory	Lung, trachea	Mild inflammation, mixed-cell infiltrate, perivascular, alveolar macrophages, alveoli/interstitium, no visible lesions
Cardiovascular	Heart, artery, aorta	Absence of necrosis/inflammatory infiltrate, mixed cells, myocardial fiber degeneration, epicardial and lesions in tissue and adventitia
Gastrointestinal	Tongue; large intestine—cecum, colon, rectum; small intestine—duodenum, ileum, jejunum	No fibrosis or visible lesions
	Esophagus	No degeneration/regeneration
	Liver	Minimal vacuolation, hepatocyte, periportal and mononuclear cell infiltration, necrosis focal/multifocal, no visible lesions
	Pancreas	No atrophy, acinar cell abnormality, no visible lesions
	Stomach	Minimal erosion, pylorus, glandular
Reproductive	Ovary, vagina, uterus/cervix, testis, epididymis	No visible lesions, cyst, squamous, absence of atrophy, degeneration in tubules
Urinary	Kidney, urinary bladder	Absence of mineralization, basophilia, dilatation, cyst in medulla, no inflammation, mixed cells in pelvis

**Table 7 pharmaceutics-17-00012-t007:** Ophthalmic evaluation of all animals during dosing and recovery phases.

Ophthalmology Parameters	Observations
Eyelids, conjunctiva cornea, lens, iris, cortex, nucleus	No observable lesions
Ophthalmoscopic (indirect ophthalmoscopy) examination using a 60-diopter lens following dilation of the pupils with 0.5% tropicamide ophthalmic solution	No abnormalities in the anterior segment of the eye
Biomicroscopic (slit lamp) examination using a Kowa SL-17 biomicroscope following dilation of the pupils with 0.5% tropicamide ophthalmic solution	No abnormalities in the posterior segment of the eye

## Data Availability

The data supporting the conclusions of this article are presented in the main text and Appendix A.

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
