# Peer review of "Evaluation of Biologics ACE2/Ang(1–7) Encapsulated in Plant Cells for FDA Approval: Safety and Toxicology Studies"

_pharmaceutics, 2024, doi:10.3390/pharmaceutics17010012_

Round 1
Reviewer 1 Report
Comments and Suggestions for Authors
In the manuscript entitled "Evaluation of biologics bioencapsulated in plant cells for FDA approval: Safety and Toxicology studies of ACE2/Ang1-7” the authors investigated Chemistry, control and manufacturing (CMC) of the ACE2/Ang (1-7) drug and the ACE2 gum. They also conducted toxicology studies using Sprague Dawley rats.
I have the following comments:
1- First of all, the topic is interesting, however the manuscript is hard to follow and needs re-editing. The manuscript
2- Figure 1: The labels of the graphs are not clear.
3- Figure 1: Figure legend should explain what is in the figure. Any abbreviations should be clarified to all readers in the legend (e.g RV: Right ventricle, LV: Left Ventricle, .... and so on)
4- Figure 1: and all figures should come after it first mentioned in the text.
5- Number of used animals: Need to be more clarified.
In abstract: n=120; 15/sex/group) in four groups
In Figure 1: as above 60 male + 60 female
However, In animal and husbandry section: 81 (plus at least 8 alternates) for each sex.
6- In drug formulation section: first and only time to mention batch: discuss batch to batch uniformity.
7- Table S1: is it a supplement? why reported in the main manuscript?
8- Table S1: confirm that if column # 6 is "Total mg Plant pow der/kg /dose" or "Total mg Plant pow der/kg /day"
9-Table S1: Justify the grouping design.
10- Table S1: The design includes placebo group 1. Why there is another placebo in group 2? why not in group 3, 4!!
11-Toxicokinetic assessment of CTB-ACE2 and CTB-Ang (1-7) section: Clarify recovery period 7 vs 20 days.
12- Same section: Read "Animals designated for terminal euthanasia (10/sex/group)". However, Table S1 mention that this group contains only male!! Clarify please.
13- Table S2: Should go to supplement
14- Figure 2: Confirm that number of animals is consistent with other locations!!
15- Table S4: move to supplement
16- Table 4: Terminal body weight provide little information about the effect of the drug. Please provide also percentage of change from original weight.
17- Table 6: Control animals potentially have underling conditions. Hence, authors have to compare groups. Mention how many animals have each pathological condition out of the total number of animals in each group.
Reviewer 2 Report
Comments and Suggestions for Authors
The manuscript submitted by Henry Daniell et al., titled “Evaluation of Biologics Bioencapsulated in Plant Cells for FDA Approval: Safety and Toxicology Studies of ACE2/Ang (1-7)”, presents a commendable and scientifically rigorous investigation. However, addressing the following critical questions and suggestions will not only elevate the overall quality of the work but also enhance its scholarly significance, ensuring it garners the recognition it deserves and is well-positioned for imminent acceptance
Detailed Comments:
- The study posits that ACE2 gum could reduce SARS-CoV-2 viral load in the oral cavity, potentially mitigating infection and transmission. The authors are encouraged to include or reference studies that evaluate the reduction of oral viral loads following ACE2 gum usage and its correlation with decreased transmission rates. In the absence of such data, a detailed outline of planned investigations to explore these aspects would enhance the study's significance.
- While the study underscores the advantages of ACE2 gum over traditional protein drug delivery systems, it lacks comparative analyses regarding cost, efficiency, and patient outcomes. The authors are advised to provide a discussion or reference studies comparing the economic and clinical benefits of ACE2 gum with conventional biologics requiring purification and cold-chain logistics. This addition would underscore its transformative potential in improving accessibility and patient compliance.
- Although the study quantifies systemic delivery of ACE2/Ang (1-7), it does not elucidate the underlying mechanisms by which oral delivery via gum ensures bioavailability and therapeutic efficacy, particularly for systemic cardiopulmonary disorders. The authors should provide a more detailed exploration of the transition of ACE2/Ang (1-7) from oral mucosa to systemic circulation, along with its anticipated therapeutic effects, to strengthen the mechanistic basis of their findings.
- The focus on ACE2 gum's application in SARS-CoV-2 is significant but overlooks its potential in other ACE2-related conditions, such as hypertension, acute respiratory distress syndrome (ARDS), or cardiovascular diseases. The authors are encouraged to expand the discussion to include the broader therapeutic implications of ACE2/Ang (1-7) gum, particularly how its bioencapsulation and non-invasive delivery could revolutionize treatment paradigms for these conditions.
- While the study presents a favorable safety profile, it lacks detailed analyses of biomarkers, organ histopathology, and specific adverse effects, particularly at higher doses. The authors should provide more comprehensive toxicology data, including biomarker profiles and detailed histological assessments, to strengthen the safety claims and support regulatory approval processes.
- Although the study establishes safety in preclinical models, it does not present sufficient evidence of clinical efficacy in humans, particularly in terms of reducing SARS-CoV-2 infection or transmission. The authors are advised to include or reference preliminary findings from human clinical studies (NCT 0543318) to demonstrate the gum's efficacy in reducing oral viral loads. If such data are not yet available, a discussion of the planned clinical endpoints and their relevance to addressing SARS-CoV-2 would add substantial value.
- The study does not address critical limitations, such as differences between rodent and human physiology or challenges in translating preclinical findings into clinical outcomes. The authors should explicitly discuss these limitations and outline strategies for addressing them in future human studies. Anticipated challenges in regulatory approval or clinical adoption should also be acknowledged, with proposed solutions to mitigate these hurdles.
- While the study demonstrates the stability of ACE2 in gum form for up to 78 weeks, it does not discuss the scalability of production or its robustness under varying storage and transportation conditions. The authors are encouraged to provide a detailed analysis of the bioencapsulation process's scalability and its feasibility for mass production. Including stability data under diverse environmental conditions would further emphasize its suitability for global distribution.
- To provide a cohesive summary of the study’s findings and their implications, the authors are strongly encouraged to include a distinct conclusion section following the discussion. This section should encapsulate the study's key outcomes, limitations, and future directions, leaving the reader with a clear understanding of its overall contribution to the field.
Reviewer 3 Report
Comments and Suggestions for Authors
Comment to paper of Henry Daniell, Geetanjali Wakade, Smruti K. Nair., et al. "Evaluation of biologics bioencapsulated in plant cells for FDA approval: Safety and Toxicology studies of ACE2/Ang1-7".
The paper is devoted to evaluation chemistry, manufacturing control, and intactness of plant cells to protect protein drugs (biologics), namely, human blood proteins ACE2/Ang1-7, in stomach including the stability, uniformity, moisture content, and bioburden. ACT2 (angiotensin converting enzyme 2) gum is the first human blood protein approved by FDA, which currently evaluated in human clinical studies to debulk SARS-CoV-2 in the oral cavity to reduce coronaviruss transmission (NCT 0543318). Manuscript provides a detailed description of GLP-compliant and IND-enabling rodent toxicology study. Safety data on oral delivery of ACE2\Ang1-7 in healthy dogs are given too. The presented data satisfied the FDA requirements for preclinical safety assessment. Chemistry, control and manufacturing studies for ACE2/Ang1-7 drug substance and ACE3 gum drug product were conducted following USP guidelines. This report lays the foundation for regulatory process approval for noninvasive and affordable human biologic drugs encapsulated in plant cells. The manuscript is good written and illustrated, and may be of interest for researchers in the field of biologics, pharmacy and pharmacology, and drug development. The paper is suitable for publication in Pharmaceutics after some corrections.
1. The Title is recommended to change as "Evaluation of biologics ACE2/Ang1-7 encapsulated in plant cells for FDA approval: Safety and Toxicology Studies".
2. There is no Conclusion part in the end of draft.
Reviewer

Round 2
Reviewer 1 Report
Comments and Suggestions for Authors
Thank you for your response to my comments. I have the following comments:
1 - Comment 16 "Table 4: Terminal body weight provide little information about the effect of the drug. Please provide also percentage of change from original weight." has not been responded to adequality in your revision as follows:
Table 4 legend read that data is "relative to start day". which is not correct. The data in the table is clearly "relative to control". A correct percentage different of body weight to start date of each group means control to start day of control, low dose to start day of low dose, and so on. This shows the safety of the treatment.
For organs, percentage difference to control is correct, but should be indicated in the legend.
The same comment is for table 4A, 5, and 5A.
2 - Line 569 reads "There were isolated organ weight values that were statistically different from their respective controls". Please, provide statistical differences in the table. e.g by star style (*, **, ***, or n.s), and indicate the used statistical test in the legend.
3- Line 576 and 581 reads "were found to be statistically insignificant (<15%)" and " exhibited statistically insignificant difference (< 10%)", respectively. % doesn't do anything with statistical significance. Dunnett's test and ANOVA provide probability. Please provide probability here instead of percentages.
Author Response
Round 2 – Reviewer 1 Response Document
Thank you for your valuable comments. Please find below the responses for each comment.
1 - Comment 16 "Table 4: Terminal body weight provide little information about the effect of the drug. Please provide also percentage of change from original weight." has not been responded to adequality in your revision as follows:
Table 4 legend read that data is "relative to start day". which is not correct. The data in the table is clearly "relative to control". A correct percentage different of body weight to start date of each group means control to start day of control, low dose to start day of low dose, and so on. This shows the safety of the treatment.
RESPONSE: The percentage change in original body weight on day 15 and day 35 has been included in tables 4, 4A, 5 and 5A for all groups and highlighted in yellow in the revised manuscript. The table legends are modified and "relative to start day" has been deleted. The details on statistical analysis and percentage difference calculation have been added to the footnotes of each table.
For organs, percentage difference to control is correct, but should be indicated in the legend.
RESPONSE: The percentage difference in organ weights relative to controls is included in the footnotes.
The same comment is for table 4A, 5, and 5A.
Response: Tables 4A, 5, and 5A have been modified accordingly.
2 - Line 569 reads "There were isolated organ weight values that were statistically different from their respective controls". Please, provide statistical differences in the table. e.g by star style (*, **, ***, or n.s), and indicate the used statistical test in the legend.
RESPONSE: Typo in line 569 has been corrected. There were NO isolated organ weight values that were statistically different from their respective controls. Statistical differences are included in tables and in the footnotes of tables 4, 4A, 5 and 5A.
3- Line 576 and 581 reads "were found to be statistically insignificant (<15%)" and " exhibited statistically insignificant difference (< 10%)", respectively. % doesn't do anything with statistical significance. Dunnett's test and ANOVA provide probability. Please provide probability here instead of percentages.
RESPONSE: Probability (p> 0.05; ANOVA and Dunnet’s Test) is included in the text and highlighted in red fonts instead of percentages.

Round 3
Reviewer 1 Report
Comments and Suggestions for Authors
Thank you for your responses to my comments.